# Alginate–Chitosan Hydrogel Formulations Sustain Baculovirus Delivery and VEGFA Expression Which Promotes Angiogenesis for Wound Dressing Applications

**DOI:** 10.3390/ph15111382

**Published:** 2022-11-10

**Authors:** Sabrina Schaly, Paromita Islam, Ahmed Abosalha, Jacqueline L. Boyajian, Dominique Shum-Tim, Satya Prakash

**Affiliations:** 1Biomedical Technology and Cell Therapy Research Laboratory, Department of Biomedical Engineering, Faculty of Medicine, McGill University, 3775 University Street, Montreal, QC H3A 2B4, Canada; 2Pharmaceutical Technology Department, Faculty of Pharmacy, Tanta University, Tanta 31111, Egypt; 3 Division of Cardiac Surgery, Royal Victoria Hospital, McGill University Health Centre, Barrie, ON L4M 6M2, Canada; 4Faculty of Medicine, McGill University, 3775 University Street, Montreal, QC H4A 3J1, Canada

**Keywords:** baculovirus, encapsulation, wound healing, angiogenesis, gene delivery, VEGFA, revascularization, wound dressing, targeted delivery, alginate, chitosan

## Abstract

Hydrogel wound dressings are effective in their ability to provide a wound-healing environment but are limited by their ability to promote later stages of revascularization. Here, a biosafe recombinant baculovirus expressing VEGFA tagged with EGFP is encapsulated in chitosan-coated alginate hydrogels using ionic cross-linking. The VEGFA, delivered by the baculovirus, significantly improves cell migration and angiogenesis to assist with the wound-healing process and revascularization. Moreover, the hydrogels have an encapsulation efficiency of 99.9%, no cytotoxicity, antimicrobial properties, good blood compatibility, promote hemostasis, and enable sustained delivery of baculoviruses over eight days. These hydrogels sustain baculovirus delivery and may have clinical implications in wound dressings or future gene therapy applications.

## 1. Introduction

Baculoviruses (BVs) are insect viruses that can be genetically engineered to express recombinant growth factors within a host. This property makes them effective for short-term gene therapy, including wound dressings [1,2,3]. Baculoviruses are also safer than alternative viral delivery systems. Specifically, many alternative viral delivery systems (lentiviral, retroviral, and adenoviral vector) are human pathogens capable of producing replication-competent viruses [1]. Moreover, the baculoviruses’ inability to integrate into the host DNA or chromosomes further supports its biosafety and use in gene delivery [2]. Baculoviruses also provide significantly higher gene expression than plasmid or other viral gene delivery due to the lack of pre-existing immunity. This is unlike adenoviruses which elicit a humoral and cellular immune response preventing re-administration [3].

Moreover, due to their high specificity, they can be genetically engineered to target only specific cell types [4]. In addition, baculoviruses can be easily scaled up, mitigating the cost typically associated with gene delivery [5]. Overall, baculoviruses provide a biosafe, low-cost, low cytotoxicity, non-replicative vector with a large cloning capacity [1,2,3]. Thus far, baculoviruses have shown promising results in treating several diseases (such as atherosclerosis, prostate cancer, and gastric cancer) in humans and animals [4,6,7]. Still, their full potential has yet to be exploited. However, some limitations must be overcome first, including serum inactivation, virus fragility, and consequential transient delivery [8,9].

Encapsulation of baculoviruses provides a promising avenue to overcome these limitations and sustain baculovirus delivery. Specifically, natural polymers such as alginate and chitosan are biocompatible and biodegradable, have extensive human safety studies, and are cost-effective options for encapsulation [10,11,12,13]. Alginate also has antimicrobial properties, can be cross-linked with calcium chloride and exhibits reversible binding to VEGF, leading to a sustained release [14]. Similarly, chitosan has anti-inflammatory, antimicrobial, and antioxidant properties [15]. Chitosan can also be complex with negatively charged particles such as baculoviruses to sustain delivery and improve transduction efficiency [16]. The polymeric capsules can also increase stability, solubility, prolong gene expression, and promote site specificity which allows for a lower dosage and less cytotoxicity [17].

Moreover, baculovirus encapsulation prevents immune inactivation, effectively shielding the virus from the immune system. Cell and bacterial encapsulation have been extensively investigated, showing promising results in localized treatment with sustained efficacy [18]. However, the efficiency and effectiveness of baculovirus encapsulation have yet to be shown and applied.

Wound healing is a complex process involving four main stages, including coagulation, inflammatory, proliferative, and remodeling phase [10]. The coagulation and inflammatory phase include platelet aggregation to help with blood clotting and removing bacteria. These platelets release growth factors to attract immune cells. The proliferative phase then commences, which includes angiogenesis, epithelialization, collagen deposition, and more. The final stage includes remodeling and when scar formation can occur. One method to enhance healing is to provide a wound dressing that provides an antimicrobial and moist environment [11]. The moist environment can promote cell migration, angiogenesis, and epithelialization leading to significantly improved healing at the injury site [19,20,21]. Specifically, hydrogels synthesized from different polymers have been shown to be effective, and some even elute growth factors, antibiotics, or pain medication to promote healing and prevent scar formation [12,13]. For instance, vascular endothelial cell growth factor A (VEGFA) is known to be one of the most potent pro-angiogenic factors and plays a key role in wound healing [22,23,24]. Specifically, VEGFA promotes endothelial cell migration, proliferation, and capillary tube formation. Furthermore, intradermal VEGF injections have accelerated diabetic wound-healing [25,26]. Here, we investigate hydrogel delivery of baculoviruses expressing VEGFA to promote cell migration and angiogenesis as a potential candidate for wound dressing applications. 

## 2. Results

### 2.1. Hydrogel Formulation Morphology

The spherical hydrogels formed instantly upon dropping the alginate virus solution into the calcium chloride cross-linking solution. The resultant alginate–BV hydrogels were coated with chitosan (termed alginate–chitosan hydrogels). The chitosan coating on the alginate capsules changed the surface morphology, as seen in Figure 1a,b. The hydrogels were observed under a brightfield microscope revealing hydrogels around 2 mm in diameter with a smooth surface appearance. The surface was more closely evaluated using FEI Quanta 450 SEM revealing a more rigid surface (Figure 1c,d). The hydrogel characteristics are outlined below in Table 1. 

The hydrogel stability was observed over several months, with most of the hydrogels remaining intact after three months at room temperature and 4 °C. However, the baculovirus diffused over time, so the hydrogels must be freshly prepared. The hydrogels swelled ~1.5-fold after initial drying in a 37 °C incubator and then slightly over time in PBS, saline, and at a pH of 4. Hydrogel swelling was observed over one month with no significant size difference over time. Moreover, the hydrogels dissolved more quickly in an acidic environment (pH of 4). 

### 2.2. Baculovirus (BV) Activity and Elution from the Hydrogels 

The supernatant from the freshly formed hydrogels was used to determine the encapsulation efficiency. The encapsulation efficiency of the hydrogels was 99.9%. Moreover, the baculovirus was released in a sustained manner over eight days. The virus activity was assessed in human umbilical vein endothelial cells (HUVECs), as seen in Figure 2. 

Both hydrogels provided sustained baculovirus elution. The alginate hydrogels had a larger initial burst release of baculovirus than those coated with chitosan. Up to 41% of cells expressed the VEGFA-EGFP when transduced with the alginate–baculovirus hydrogels on day 4. Comparatively, up to 40% of cells expressed the gene of interest when transduced with the baculovirus eluted from the alginate–chitosan hydrogels on day 8. The virus elution also remained stable over eight days for the alginate hydrogels and over 10 days for the alginate–chitosan hydrogels. The baculovirus elution slowly diminishes over time. Hydrogel delivery improves the therapeutic efficacy by releasing the baculovirus over several days instead and all at once if left unencapsulated. The baculovirus VEGFA delivery increased VEGFA expression up to 25.76-fold with an MOI of 50 compared to the mock-infected cells (PBS).

### 2.3. Hydrogel Formulations Delivering Baculovirus Demonstrate Therapeutic Potential over 7 Days

The baculovirus eluted from the hydrogels maintained activity. The baculovirus-hydrogel system demonstrated beneficial properties, including no cytotoxicity, good blood compatibility, antimicrobial properties, and significantly improved wound healing and angiogenesis compared to the mock cells (Figure 3a–g). VEGFA activity was maintained from the baculovirus, as demonstrated by the improved cell migration over time. Both hydrogels significantly improved cell migration in HUVECs over the course of seven days (Figure 3c–h). Moreover, the alginate hydrogels demonstrated significantly less cell migration when compared to chitosan-coated hydrogels. However, the alginate hydrogels were still significantly better at promoting wound healing compared to the mock-infected cells (Figure 3c). Angiogenesis also significantly improved with the baculovirus treatment, as shown by the longer branching and increased number of nodes after tube formation (Figure 3a,b). The hydrogels also prevented *E. coli* and *C. albicans* growth directly below and inhibited *C. albicans* growth surrounding the alginate–chitosan hydrogels (Figure 3i). 

The baculovirus delivery system also demonstrated a high safety profile. Specifically, there was no significant decrease in proliferation or viability in the hydrogel-treated cells compared to the mock-infected cells. The baculovirus significantly improved cell proliferation compared to the mock transduced cells (Figure 4a). The viability also remained high for all treatment groups, above 94.3%, with no significant difference except for the alginate hydrogels (Figure 4b). Moreover, the baculovirus and hydrogel delivery system demonstrated blood biocompatibility with no significant hemolysis (Figure 4c). The baculovirus, gene expression, and hydrogel formulations all showed negligible hemolysis, below 5%. The hydrogels also promoted whole blood clotting over time compared to the polystyrene “mock” control (Figure 4d). Overall, the baculovirus-eluting hydrogels demonstrated no cytotoxicity or antimicrobial properties, and improved cell migration and angiogenesis. 

## 3. Discussion

The hydrogel delivery formulation presented here is cost-effective, requiring few resources and natural polymers (alginate and chitosan) that are widely abundant and inexpensive. Alginate and chitosan are also known for their excellent biocompatibility, bioadhesion, and biodegradability [27]. Moreover, the hydrogels do not require harsh solvents, promoting their safety and preventing baculovirus inactivation during synthesis. The hydrogels are also easily pre-sterilized to avoid bacterial contamination in the wound. 

The ionically cross-linked hydrogels were smooth and demonstrated good stability at various temperatures and osmotic gradients. Their stability at different temperatures is also beneficial to improve the feasibility and allows for sufficient storage time before expiration. Moreover, their stability in saline, PBS, and at a pH of 4 demonstrates their ability to perform in various physiological conditions. A pH of 4 would be comparable to the acidic environment observed as wounds begin to heal [28]. Moreover, a 1% concentration of acetic acid has been shown to be effective against a wide range of bacteria and fungi, effectively preventing infection and multi-resistant bacteria [29]. Beneficially, the chitosan coating is dissolved in acetic acid, which creates an acidic environment to promote wound healing [28]. Moreover, hydrogel swelling is important for bacterial entrapment and microbial growth inhibition [19]. 

The high baculovirus encapsulation (99.9%) promotes the system’s deliverability and minimizes costs with the little virus going to waste. The baculovirus elution was sustained over eight days, overcoming the transient baculovirus delivery previously reported [8,9]. Moreover, sustained baculovirus elution has a vast potential in chronic wound environments, whereby healing is a slow process. The virus activity also remained active and was capable of transducing 40% of cells with an MOI of 50. Overall, this delivery system can prolong baculovirus delivery with a therapeutic effect. 

The baculovirus eluted from the hydrogels also possessed beneficial properties for wound-healing applications, including no cytotoxicity, good blood compatibility, and significantly improved wound healing and angiogenesis compared to the mock cells. Specifically, there was no significant difference in proliferation or viability in the hydrogel-treated cells compared to the mock-infected cells. Moreover, the baculovirus and hydrogel delivery system demonstrated blood biocompatibility with no significant hemolysis. The hydrogels also promoted blood coagulation over time (15–45 min), effectively preventing excessive bleeding and creating a barrier for the wound. The VEGFA expression from the baculovirus hydrogels also significantly improved HUVEC migration over eight days leading to improved considerably wound healing.

Moreover, the baculovirus delivery significantly improved tube formation in the angiogenesis assay, which can assist with wound revascularization. The chitosan coating demonstrated the best wound-healing ability, which may be attributed to improved baculovirus transduction due to the positively charged chitosan and slightly acidic pH created by the polymer coating [20]. Overall, the chitosan coating prolonged baculovirus delivery and improved wound healing, cell proliferation, and antimicrobial resistance. 

Some limitations include investigating the immune and cytokine response and other growth factors. The possibility of fibromatosis due to growth factor elution should also be investigated. However, preliminary studies reveal that VEGF promotes fibrogenesis but is also needed for fibrosis resolution [21]. Future studies should also investigate the in vivo impact on healing with reference to the immune response and the several organ systems at play. Overall, these baculovirus-eluting hydrogels enable sustained baculovirus delivery which releases VEGFA to promote cell migration, proliferation, and tube formation along with beneficial safety properties in vitro. The baculovirus-eluting hydrogels may have future applications in wound healing through the promotion of angiogenesis. 

## 4. Materials and Methods

### 4.1. Insect Cell Culture 

Sf21insect cells (Invitrogen Life Technologies, Carlsbad, CA, USA) were maintained at 27 °C in SF900 II serum-free medium in T-75 flasks or 250 mL shake flasks (Erlenmeyer, CA, USA). The Sf21 cells in the shake flasks were maintained in a shaking incubator at 130 rpm. The cells were subcultured 2–3 times per week to maintain the exponential growth phase.

### 4.2. Gene Cloning 

The human VEGFA in a pcDNA3.1 + eGFP vector was purchased from GenScript. The EGFP-tagged VEGFA gene was excised from the original plasmid using EcoRI and XbaI Fast Digest restriction enzymes (Thermo Fisher, Waltham, MA, USA). The resultant genes of interest were run on a 1% (*m*/*v*) agarose gel containing SYBR Green (Thermo Fisher) at 100 Volts. After one hour, the DNA bands were visualized using a blue light transilluminator (miniPCR bio, Boston, MA, USA). Each gene fragment was excised and purified using NEB’s Gel Extraction Kit. The genes of interest were each ligated, using Instant Sticky End Ligase (NEB), into the pOET6 plasmid with compatible sticky ends. The VEGFA-eGFP-pOET6 plasmid was then chemically transformed into DH5alpha *E. coli* (Thermo Fisher) and selected for using the ampicillin resistance present within the pOET6 plasmid (MJS BioLynx Inc., Brockville, ON, Canada). Several different LB agar plates were streaked with different dilutions of the transformed bacteria and grown overnight. The next day, the resistant colonies containing the gene of interest were selected and grown overnight to amplify the plasmid. The plasmid was then extracted and purified using NEB’s plasmid purification kit. The resultant purified pOET6 plasmid, each with a gene of interest, was used for all future virus production steps or frozen at −20 °C for future use. 

### 4.3. Baculovirus Production

For baculovirus production, the supplier’s protocol was followed [30]. In an exponential growth phase, 5 × 10^5^ Sf21 cells were seeded into a 12-well plate 1 h before virus transfection. Next, 200 ng of VEGFA-EGFP-pOET6 plasmid was added to 100 ng of flashBAC DNA (Oxford Expression Technologies, Oxford, UK), 0.48 μL of TransIT Insect Reagent (MJS BioLynx, Brockville, ON, Canada), and 100 μL PBS and incubated at room temperature for 15 min. The transfection mixture was then added to the Sf21 cells and incubated overnight at 27^o^C. The next day, 0.5 mL of SFM-II was added. Five days after transfection, the culture medium was harvested and centrifuged at 1000× *g* for 10 min. The supernatant was collected and stored at 4 °C (P_o_ virus stock). To amplify the baculovirus, 100 mL of Sf21 cells were diluted to 2 × 10^6^ cells/mL in SF900II medium, and 0.4 mL of P_o_ virus stock was added. The infected Sf21 cells were agitated at 130 rpm for four days before harvesting the culture medium as described above. This generated P_1_ baculovirus stock expressing VEGFA. 

### 4.4. Baculovirus Titration 

The baculovirus was titrated using viral plaque assays or endpoint dilutions to determine the initial stock concentration and encapsulation efficiency, respectively. For the plaque assay, 0.5 × 0^6^ cells were seeded into each well of a 12-well plate and incubated for 1 h. A sample of the P_1_ virus stock was used to make serial dilution (down to 10^−7^). After the one-hour incubation, the media was removed from each well and 100 μL of each dilution (10^−4^ to 10^−7^ dilutions) was added to a well in triplicate. The infected Sf9 cells were incubated at room temperature for 1 h, after which the 100 μL of inoculum was removed. For the plaque assay, a 1% (*m*/*v*) agarose-SFM-II overlay was added to the side of each well. The overlay solidified at room temperature, and 0.5 mL of SFM-II was added. The plate was then incubated at 27 °C for 4 days. After 4 days, 0.5 mL of neutral red (0.25 mg/mL) was added to each well and incubated for 3 h. After 3 h, the neutral red stain was removed, and the plates were inverted to allow the plaques to clear. Wells with 10 to 30 plaque-forming units (PFUs) were counted, and the plaque count was averaged to determine the viral titer using the following equation.
Titer of virus (pfu/mL) = (average plaque count) × (dilution factor *) × (10 **).* Dilution factor = the inverse of the dilution used on the counted plate.** Multiply by 10 because 0.1 mL was applied to each dish.

For the endpoint dilution, 10,000 Sf21 cells were seeded per well in a 96-well plate and incubated for 1 h. During this time, serial dilutions of the virus were made using SFM-II (down to 10^−10^). After the 1 h incubation, each virus dilution was added to 10 separate wells. The plate was then incubated for five days and observed for cytopathic effect (CPE) in each well. The titer was calculated using the following equation.
Titer (pfu/mL) = 10(1 + Z (X − 0.5))
where Z is Log 10 of the starting dilution (1 for a ten-fold dilution) and X is the sum of the fractions of CPE-positive wells.

### 4.5. Endothelial Cell Culture 

Human umbilical vein endothelial cells (HUVECs) were purchased from Sigma Aldrich. The HUVECs were maintained in T-25 flasks in a 37 °C, 5% CO_2_ incubator. A complete endothelial growth medium (Sigma Aldrich, St. Louis, MI, USA) was used to culture the HUVECs. Cells were used within 4–5 passages upon receiving.

### 4.6. Spherical Hydrogel Preparation

Ionic cross-linking was employed for virus encapsulation under aseptic conditions. First, a 4% (*m*/*v*) sodium alginate, 2% (*m*/*v*) chitosan with 0.5% (*v/v*) acetic acid, and 100 mM calcium chloride solution was prepared (Sigma Aldrich and Bio-Basic, Markham, ON, Canada). The sodium alginate and chitosan were sterilized in the autoclave for 20 min. The calcium chloride was sterilized using a 0.20 μm filter. Next, 1.5 mL of sodium alginate solution was mixed with either sterile water for the mock hydrogels or baculovirus stock in a 50:50 ratio. The remaining 2% (*m*/*v*) alginate solution was then loaded into a 5 mL syringe and added dropwise, using a 30 gauge needle, into a beaker containing 20 mL of calcium chloride, stirring on the lowest possible setting. The hydrogels formed instantly and were allowed to solidify in the beaker for 10 min. After this, the calcium chloride solution was aspirated off, and 1 mL of the 2% (*m*/*v*) chitosan solution was added for 5 min. The supernatant was saved to determine the encapsulation efficiency. The hydrogels were washed with PBS three times and finally resuspended in PBS to evaluate the virus elution over time. 

### 4.7. Hydrogel Morphology and Swelling

To observe the morphology, freshly formed hydrogels were imaged under an inverted microscope using brightfield at 4× magnification. The size of 10 random hydrogels was measured using ImageJ. Some of the spherical hydrogels were dried overnight, imaged, and measured to determine the swelling ratio using the following equation: Swelling ratio= Weight_swollen_/Weight_dry_

### 4.8. Scanning Electron Microscopy (SEM)

The uncoated and coated hydrogels were removed from suspension and dried overnight in a 37 °C incubator. They were then visualized using a FEI 450 Quanta SEM under high vacuum at 10 kV. 

### 4.9. Zeta Potential 

The hydrogels were resuspended in MilliQ water. Next, a zeta potential analyzer with electrophoretic laser Doppler anemometry (Brookhaven Instruments Corporation, Holtsville, New York, NY, USA) was used to determine the surface charge of the hydrogels. Zeta Potential Analyzer version 3.57 software was used to determine the zeta potential. Each measurement was obtained after taking the average of the ten runs. 

### 4.10. Encapsulation Efficiency of the Hydrogel Delivery Formulation

The supernatant from the virus encapsulation process was preserved and titrated to determine the encapsulation efficiency.
Encapsulation efficiency % = encapsulated virus concentration/ initial virus concentration × 100%

### 4.11. Stability

The stability of the hydrogels was evaluated using an osmotic pressure and a rotational stress test. The osmotic pressure test was performed by transferring 50 microcapsules into deionized water in a flask. The rotational stress test consisted of transferring 50 microcapsules into a hypotonic solution (deionized water) and shaking it at 150 rpm at 37 °C for several days. The hydrogels were also resuspended in PBS pH 7.4, saline solution, or PBS pH 4 (addition of HCl) and shaken to test their stability in a variety of environments. 

### 4.12. Antimicrobial Studies 

The antimicrobial properties of the alginate and alginate–chitosan hydrogels were tested using an adapted disc diffusion method [31]. Briefly, *E. coli* was grown in Luria-Bertani (LB) medium, and *C. albicans* were grown in YPD media. Dilutions of the bacteria were spread on LB agar or YPD agar plates to determine the number of colony-forming units (CFU). The bacteria were diluted to 10^8^ CFU/mL and 100 μL was spread onto fresh LB/YPD agar plates. The different hydrogels were plated onto the plates containing *E. coli* or *C. albicans* and incubated at 37 °C for up to 24 h. A drop (10 μL) of penicillin was used as a negative control. After 24 h, the area of growth inhibition was observed and measured. 

### 4.13. Baculovirus Release and Activity 

Release studies were performed by incubating freshly formed hydrogels, containing baculovirus in 2 mL of PBS. The supernatant (baculovirus in PBS) was removed and replaced with fresh PBS every 24 h until all virus was released. The hydrogel suspension media was then used to transduce mammalian cells (HUVECs) in 96-well plates to determine the virus activity. 

### 4.14. Baculovirus Transduction 

The hydrogel suspension media or the virus stock itself was added to either HUVECs, in black 96-well plates (Corning 3603) and incubated at 37 °C. After 3 h of incubation, the virus or hydrogel suspension media was removed and replaced with fresh cell media. The cells were incubated for 24–48 h to allow for viral gene expression. These cells were then imaged using the green and blue fluorescent filter in the ImageXpress Micro^®^ Confocal High-Content Imaging System (Molecular Devices, San Jose, CA, USA). At 24 h, the cells were imaged using brightfield and fluorescent microscopy (green filter). At 48 h, the percent of fluorescent cells was compared to the number of cells using DAPI staining for the nuclei (blue filter) and a green filter for the EGFP-tagged VEGFA expression. 

### 4.15. Baculovirus Gene Expression

VEGFA PCR primers were purchased from Bio-Rad. HUVECs (2 × 10^4^ cells/well) were seeded into a 48-well plate and incubated overnight. The HUVECs were then transduced with different MOIs of baculovirus for 3 h after which the virus inoculum was removed and replaced with fresh cell media. At 24 h post-infection (hpi), the media was saved for the angiogenesis assay, and the total RNA from each well was extracted using Bio-Basic’s Total RNA Extraction Kit. The extracted RNA was then mixed with the Luna Universal One-Step RT-qPCR kit (NEB) and primers. The mixture was then placed into the Eco Illumina PCR^®^ system. Amplification was carried out for 40 cycles with 35 s (denaturation), 55 °C for 35 s (annealing), and 72 °C for 25 s (extension). 

### 4.16. MTT Proliferation Assay 

A MTT cell growth assay kit was purchased from Sigma Aldrich. As above, different MOIs of BV and the hydrogel supernatant were added to HUVECs (10,000 cells/well in a 96-well plate) and incubated for 3 h. PBS and DMSO was used as positive and negative control, respectively. After the 3 h incubation, the baculovirus supernatant was removed and replaced with fresh media. Next, 24 hpi, 0.01 mL of AB Solution (MTT) was added to each well. The cells were incubated for 4 h at 37 °C to allow MTT cleavage to occur. After one hour, 0.1 mL of isopropanol with 0.04 N HCl was added to each well. The isopropanol solution was mixed thoroughly via pipetting. The plate was then read using an EnSpire Multimode plate reader (Perkin Elmer, Waltham, MA, USA) with a wavelength of 570 nm. 

### 4.17. Live/Dead Assay

To estimate the cytotoxicity of the hydrogels, a Live/Dead Assay was employed. First, 2 × 10^4^ HUVECs/well were seeded into a 48-well plate and incubated overnight. The next day, the hydrogels were added to each well in triplicate and incubated. Media alone or DMSO was used as a control. After 24 h, 5 μL of 1 mM Calcein AM and 5 μL of 2.5 mg/mL propidium iodide (both from Thermo Fisher) were added to 10 mL of cell media. This solution was then added to the cells in the 48-well plate and incubated at 37 °C for 30 min. After 30 min, the cells were imaged (three random images per well) using the green and red filters on the Leica DMIL microscope with a Canon T3i camera. ImageJ ‘analyze particles’ was used to count the number of live (green) and dead (red) cells.

### 4.18. Hemocompatibility 

Blood samples from two individuals were obtained from Innovative Research and tested independently in triplicate. The potential hemolysis of the virus itself, gene expression, and the hydrogels were evaluated. Briefly, all tested samples were immersed into 5 mL of PBS in a 15 mL centrifuge tube. Next, 4 mL of citrated blood was mixed with 5 mL PBS and 0.1 mL of the diluted blood was added to each sample. The samples were incubated at 37 °C for 1 h and then centrifuged at 1000 rpm for 10 min. The supernatant containing the lysed hemoglobin was placed into a 96-well plate, and the absorbance was read at 545 nm. The negative and positive control was PBS and deionized water, respectively. The following equation was used to determine the % hemolysis.
Hemolysis (%) = ((OD_sample_ − OD_negative control_)/(OD_positive_ − OD_negative_)) × 100%

Blood clotting on the hydrogels was also evaluated using a method described by Sabino et al. (2020) [22]. Briefly, the hydrogels were placed into a 24-well plate. Next, 7 μL of whole blood was pipetted onto the surface of the hydrogels. At 15, 30 and 45 min, deionized water was added to each sample and incubated for 5 min. The resulting supernatant was removed, and the absorbance was read on a 96-well plate at 540 nm. As a control, 7 μL of whole blood in distilled water was used as a control where no blood clotted onto a biomaterial surface. A ‘mock’ polystyrene surface was also used to compare the blood coagulation properties of the hydrogels. 

### 4.19. Wound-Healing Assay

Growth factors, such as VEGFA, induce endothelial cell migration, as demonstrated by a wound-healing or scratch assay [23]. To confirm the gene activity eluted from the virus, HUVECs were seeded in 96-well plates and grown to confluency. Twenty-four hours prior to wound healing, the cells were transduced. Specifically, the wells were incubated with baculovirus for 2 h (PBS was used as a mock), and then the virus solution was replaced with fresh cell media. The next day, a straight wound was generated down the middle of each well using a 200 μL pipette tip. After 12 h, the cells were visualized and imaged using a Leica DMIL microscope with a Canon T3i camera. After 24 h, the cells were fixed with 4% (*w/v*) paraformaldehyde and stained with crystal violet (Thermo Fisher). The wound-healing ability was evaluated using an open-access ImageJ plugin [24].

### 4.20. Endothelial Tube Formation Assay

To confirm the angiogenic potential of VEGFA released from the baculovirus a standard angiogenesis assay was performed as described elsewhere with slight modifications [24]. First, the supernatant from baculovirus transduced cells (containing the eluted VEGFA) was added as the angiogenic stimulator. Endothelial cell media with VEGF was used as a positive control, and DMSO (a known inhibitor of angiogenesis) was used as the negative control [32]. The cells were then incubated until tube formation was observed. After 2 h, the wells were imaged using a phase contrast microscope. The images were analyzed using an ImageJ plugin [33]. 

### 4.21. Statistical Analysis

All experiments were performed at least twice on different days and with three replicates each time. The values are expressed as the mean ± standard deviation for each experiment. SPSS (SPSS Inc., Chicago, IL, USA, IBM version 28) was used to perform all statistical analyses and ImageJ was used for all image analyses. A *p*-value of less than 0.05 was considered statistically significant. 

## 5. Conclusions

Baculovirus’ VEGFA gene delivery to endothelial cells is efficient and safe. Alginate and chitosan–alginate hydrogels, with an encapsulation efficiency of 99.9%, can prolong virus delivery and increase the therapeutic window. Moreover, the encapsulated baculovirus maintained its activity and was released in a sustained manner over eight days. The virus transduction efficiency can reach over 40% with the hydrogel delivery system. The baculovirus elution stimulates tube formation, cell proliferation, and cell migration, all of which contribute to angiogenesis. This may be beneficial for chronic wound-healing applications. Moreover, this is a proof of concept that encapsulation can prolong baculovirus elution, making it a candidate for several gene therapy applications. 

## Figures and Tables

**Figure 1 pharmaceuticals-15-01382-f001:**
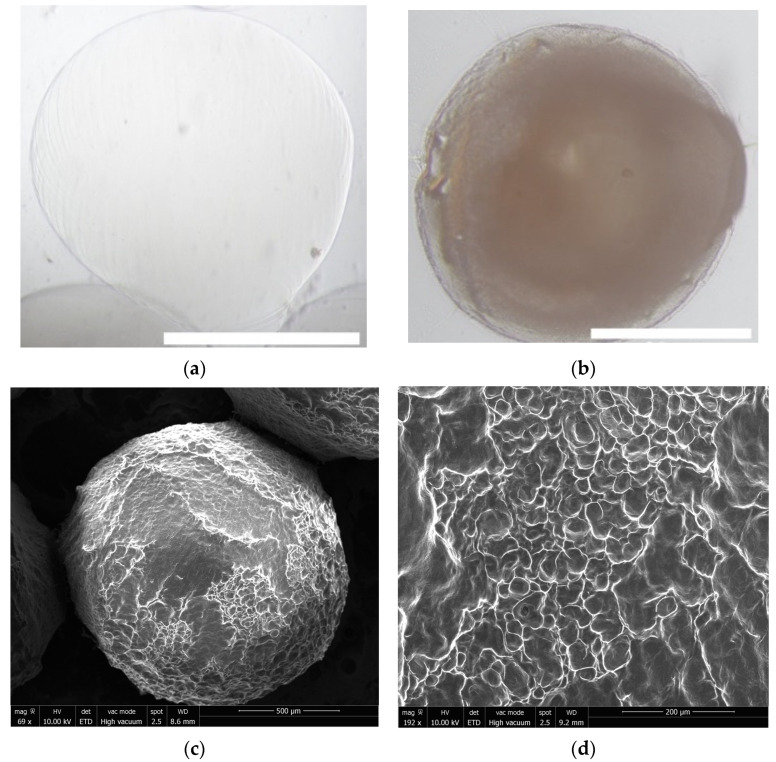
Brightfield images of the hydrogels swollen in PBS at 4×, the scale bar represents 1 mm. (**a**) Alginate hydrogel; (**b**) alginate hydrogel coated with chitosan; (**c**,**d**) SEM images of the dried hydrogels at different magnifications.

**Figure 2 pharmaceuticals-15-01382-f002:**
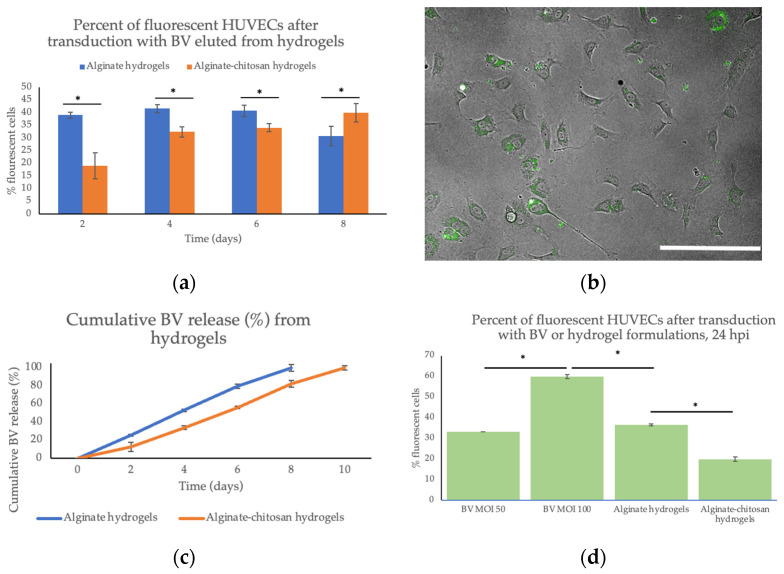
(**a**) Percentage of fluorescent HUVECs after transduction with baculovirus eluted from hydrogels. (**b**) Brightfield image overlaid with a fluorescent image of HUVECs transduced with baculovirus eluted from alginate–chitosan hydrogels on day 3 at 20× magnification, scale bar = 200 μm. (**c**) Cumulative BV release (%) from alginate and alginate–chitosan hydrogels. (**d**) Percentage of fluorescent HUVECs after transduction with BVs or hydrogel suspension media, 24 hpi. *n* = 3 per group for each experiment. * Indicates a significance of *p* < 0.05.

**Figure 3 pharmaceuticals-15-01382-f003:**
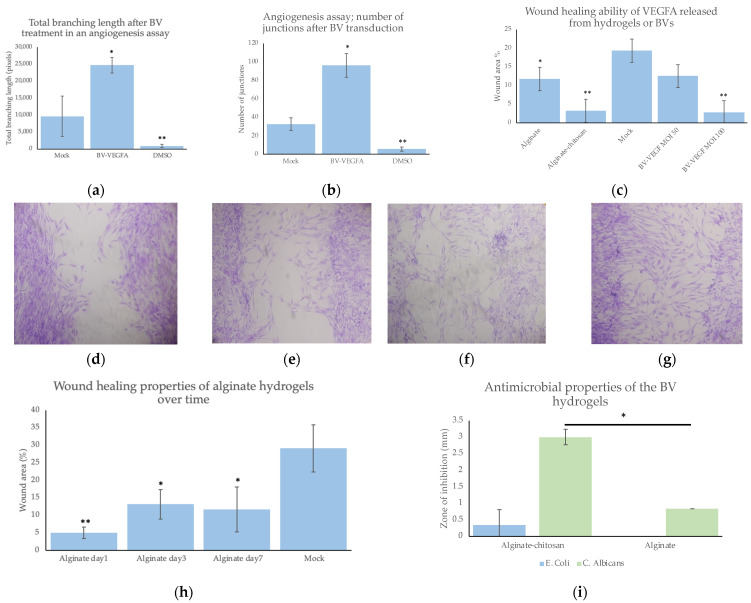
(**a**) Total branching length of tube formation after treatment with BV expressing VEGFA compared to the mock transduced cells. (**b**) A number of junctions formed during angiogenesis assay compared to the mock transduced cells. (**c**) Wound area remaining after hydrogel or baculovirus treatment for 12 h compared to the mock transduced cells. (**d**–**g**) Brightfield images of wound area 12 h after wounding and staining with crystal violet at 10×; (**c**) mock transduced and (**d**) transduced with BV released from alginate hydrogels on day 8; (**e**) transduced with BV released from chitosan–alginate hydrogels on day 8; (**h**) control baculovirus transduction with a MOI of 100; (**g**) wound area (%) after hydrogel treatment over time compared to the mock cells. (**i**) Growth inhibition zone of bacteria from alginate and alginate–chitosan hydrogels using the antimicrobial disc diffusion method. *n* = 3 per group. * Indicates a significance of *p* < 0.05 and ** indicates a significance of *p* < 0.005.

**Figure 4 pharmaceuticals-15-01382-f004:**
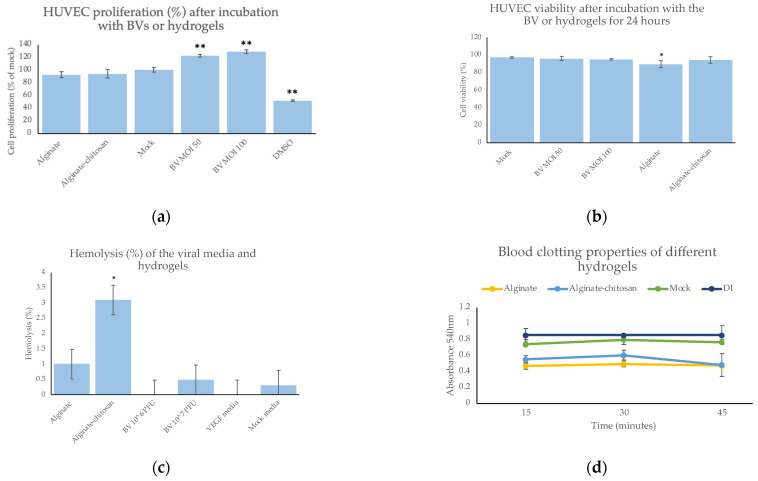
(**a**) MTT proliferation assay with HUVECs after treatment with different hydrogels or the BV compared to the mock transduced cells. (**b**) HUVEC viability (%) after BV transduction or hydrogel incubation for 24 h compared to the mock cells. (**c**) Compared to the mock cell media, hemolysis (%) of the hydrogels, baculovirus (BV), and VEGFA expressed from the BV. (**d**) Whole blood clotting on the surface of the hydrogels compared to polystyrene (mock) and deionized water (control). *n* = 3 per group and repeated twice. * indicates *p* < 0.05 and ** indicates *p* < 0.005.

**Table 1 pharmaceuticals-15-01382-t001:** The properties of the hydrogel formulations. The zeta potential of the hydrogels, after being washed and resuspended in Milli-Q water (*n* = 3/group, 10 runs each). The average size of 10 randomly selected spherical hydrogels. The average swelling ratio of the different hydrogel formulations (10 different hydrogels/group, selected at random).

Property	Alginate	Alginate–BV	Alginate–Chitosan	Alginate–Chitosan–BV
Zeta potential (mV ± SD)	−2.67 ± 0.87	−2.85 ± 1.66	6.62 ± 0.35	−0.13 ± 0.42
Size (mm ± SD)	2.17 ± 0.06	2.12 ± 0.07	2.21 ± 0.10	2.37 ± 0.10
Swelling ratio (average ± SD)	1.51 ± 0.05	1.48 ± 0.05	1.54 ± 0.07	1.65 ± 0.07

## Data Availability

Data is contained within the article.

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
