# Peer review of "Alginate–Chitosan Hydrogel Formulations Sustain Baculovirus Delivery and VEGFA Expression Which Promotes Angiogenesis for Wound Dressing Applications"

_pharmaceuticals, 2022, doi:10.3390/ph15111382_

Round 1

Reviewer 1 Report

Schaly et al. describes a biopolymer-based microparticle as a gene delivery system. Even though, alginate and chitosan are one of the most reported polysaccharides used in microcapsules preparation, little has been reported in the use of these systems to deliver a baculovirus.

The strength of this work is the successful synthesis of these microcapsules with an encapsulation efficiency of ~100%. Their transduction capacity was evaluated, just like their cytotoxicity (in HUVECS and SMC) and biocompatibility (in human red blood cells), as well as their antimicrobial properties.

Will like to point out that this manuscript is well organized. The points below should be considered by the authors to improve their manuscript.

1-Introduction.

1.      The lack of references:

·         Line 30-33

·         Line 49

·         Line 53-54

2.      Reference 6 line 43, does not seem to be appropriate/correct

3.      What are the possible drawbacks of this baculovirus technology?

4.      It is not well defended why VEGF-A can promote cell migration and angiogenesis. A brief explanation should be given.

5.      Prolonged delivery of an excessive dose of a growth factor, fibromatosis could be a concern??

2- Lack of precision.

1.      In all the document ml should be replaced by mL ou μl to μL.

2.      The scale does not add up. So, in figure 1 by SEM the microparticles are bigger then 1 mm, while of the brightfield images they are around 200 μm.

3.      What is the voltage used in the electrophorese. (line200).

4.      What is the reference of the cell culture medium used (line253)

5.      In the preparation of the microparticles indication of the solution of the polymers are referred in percentage, indication of the physical phase needs to be mentioned eg. %(v/v) or %(m/v) (line259).

6.      How was the transduction of cells (SMC or HUVECs) preformed.

7.      “toxicity” should be cytotoxicity within the whole manuscript.

8.      Proliferation assay should be described.

3-Explanaition.

1.      Figure 2a . A comparison between the normal transduction method and theses microparticles need to me shown.

2.      Figure 2c and d. Is their no SD? No description of the N value is given in the caption.

3.      Where are the results to back up the “compared to baculovirus transduction alone which lasts up to 2 days” (line99).

4.      Please check line 108-109 with line 111-112.

5.      Antibacterial results must be presented.

6.      Cytotoxicity results should be presented.

7.      Hemolysis results should be presented.

8.      Representative wound-healing images from the wound-healing assay overtime for each condition should presented.

9.      A proper angiogenesis assay should be presented to backup the statement indicated in line178.

Author Response

Overall response: 

We would like to thank you for your valuable suggestions and comments. In the revised manuscript, we have addressed those comments and suggestions carefully. The followings are the detailed of responses to your specific comments together with the specific changes that we have made to the revised manuscript.

Comments:

Schaly et al. describes a biopolymer-based microparticle as a gene delivery system. Even though, alginate and chitosan are one of the most reported polysaccharides used in microcapsules preparation, little has been reported in the use of these systems to deliver a baculovirus.

The strength of this work is the successful synthesis of these microcapsules with an encapsulation efficiency of ~100%. Their transduction capacity was evaluated, just like their cytotoxicity (in HUVECS and SMC) and biocompatibility (in human red blood cells), as well as their antimicrobial properties.

Will like to point out that this manuscript is well organized. The points below should be considered by the authors to improve their manuscript.

1-Introduction.

  1. The lack of references: Line 30-33, Line 49, Line 53-54

Response: Thank you. References have been added to line 30-33, 49, and 53-54.

Line 53-54 has been moved to the discussion section because this is what the paper hopes to show.

  1. Reference 6 line 43, does not seem to be appropriate/correct

Response: You are correct, reference 5 (previously #6) has been updated. Thank you.

  1. What are the possible drawbacks of this baculovirus technology?

Response: Possible drawbacks of baculovirus gene delivery has been added to line 40 with references 14-15. A limitations section has also been added to the discussion line 358-365.

  1. It is not well defended why VEGF-A can promote cell migration and angiogenesis. A brief explanation should be given.

Response: An explanation on the pro-angiogenic potential of VEGFA has been included in line 143-147 with references 28-31 to support this conclusion.

  1. Prolonged delivery of an excessive dose of a growth factor, fibromatosis could be a concern??

Response: A limitations section has also been added to the discussion, line 358-365, with concern for fibromatosis added. Reference 37 is also used in support.

2-Lack of Precision

  1. In all the document ml should be replaced by mL or μl to μL.

Response: Thank you for pointing this out, it has been updated throughout. For example, line 371.

  1. The scale does not add up. So, in figure 1 by SEM the microparticles are bigger then 1 mm, while of the brightfield images they are around 200 μm.

Response: Thank you for pointing this out, I had the incorrect scale on my brightfield images. This has been corrected. The microparticles under SEM are around half the size due to the drying which has been included in the Figure caption.

  1. What is the voltage used in the electrophorese. (line200).

Response: The voltage has been added into line 205, thank you.

  1. What is the reference of the cell culture medium used (line253)

Response: The supplier of the cell culture medium and supplements has been added. Thank you.

  1. In the preparation of the microparticles indication of the solution of the polymers are referred in percentage, indication of the physical phase needs to be mentioned eg. %(v/v) or %(m/v) (line259).

Response: Thank you, it has been added to all the polymer solution references.

  1. How was the transduction of cells (SMC or HUVECs) preformed.

The baculovirus transduction method has been split into its own heading under 4.14.

  1. “toxicity” should be cytotoxicity within the whole manuscript.

Response: Thank you. It has been updated throughout. For example, in like 16, 36, 127, ….

  1. Proliferation assay should be described.

Response: The proliferation assay is reference to the MTT assay. We have updated the name to an MTT proliferation assay throughout the manuscript.  A Live Dead assay has also been added.

3-Explanation

  1. Figure 2a. A comparison between the normal transduction method and theses microparticles need to be shown.

Response: Normal transduction values 24 hours after infection with different MOIs have been added to Figure 2d.

  1. Figure 2c and d. Is there no SD? No description of the N value is given in the caption.

Response: SD values and the N value has been added to the Figure, thank you.

  1. Where are the results to back up the “compared to baculovirus transduction alone which lasts up to 2 days” (line99).

Response: We meant to say the microparticles sustained baculovirus elution which when compared to adding baculovirus directly to cells lasted over 8 days. This has been corrected please see line 200-203.

  1. Please check line 108-109 with line 111-112.

Response: Thank you, this was a typo that has been corrected.

  1. Antibacterial results must be presented.

Response: Antibacterial results have been added to Figure 3h.

  1. Cytotoxicity results should be presented

Response: Cytotoxicity has been renamed to MTT proliferation assay and is shown in Figure 4a. The cell viability is also shown in Figure 4b.

  1. Hemolysis results should be presented.

Response: Hemolysis results have been added to Figure 4c.

  1. Representative wound-healing images from the wound-healing assay overtime for each condition should presented.

Response: Wound healing has been evaluated and presented in Figure 4g. Example images are shown in Figure c-f for day 8.

  1. A proper angiogenesis assay should be presented to backup the statement indicated in line178.

Response: The angiogenesis assay method has been updated and results presented. The assay supports that the VEGFA release from the BV (encapsulated in the MPs) is functional and significantly improves tube formation when compared to cells treated with media containing VEGF and cells treated with DMSO (an angiogenesis inhibitor).

Reviewer 2 Report

In this paper, the authors introduced microparticles system based on alginate-chitosan for sustaining baculovirus VEGFA delivery for wound healing applications. This is interesting; however, this paper lacks novelty, and the provided data is not enough to support the conclusion, such as 1) the microparticles were not quantitatively investigated; 2) there were not SD values for the release of BV (FIGURE 2); 3) in vitro and in vivo data are not enough, photos should be given, and so on. 

Author Response

We would like to thank you for your valuable suggestions and comments. In the revised manuscript, we have addressed those comments and suggestions carefully. The followings are the detailed of responses to your specific comments together with the specific changes that we have made to the revised manuscript.

  1. The microparticles were not quantitatively investigated.

Response: The average size, swelling ratio, and zeta potential of the microparticles is given in Table 1. Thank you for your suggestion.

  1. There were not SD values for the release of BV (Figure 2)

Response: Thank you, SD values have been added to the BV release graph. The BV release has also been combined into one singular graph.

  1. In vitro and in vivo data are not enough, photos should be given and so on.

Response: More data has been added. For example, the hemolysis, MTT proliferation, viability (from a Live Dead assay), antibacterial, zeta potential, swelling ratio, baculovirus transduction compared to the MPs, and microparticle (MP) size has been included. This is in addition to the brightfield images, fluorescent images, blood coagulation properties of the microparticles, wound healing/scratch assay, and tube formation assay. Future studies would work on evaluating this system in vivo and has been added to the discussion. We have also restructured the paper so that all conclusions are supported by results. We have also made the paper more specific to show that the main finding is that baculovirus delivery can be prolonged and that it could have potential applications in wound dressings.

Round 2

Reviewer 1 Report

My only concern is regarding the captions of the figures. They should have more description or even the figures that they describe E.g. Figure 1. Brightfield images of swollen microparticles at 4x, the bar represent 1 mm. a) alginate b) alginate coated with chitosan c-d) SEM image of dried microparticles. e) Zeta potential of the BV and MPs. Where is the zeta image? Description for the zeta N=? (how many replicate?) E.g. Table 1. Microparticle properties. What is discribed in the table? Zeta, hydrodynamic size (by DLS?) N=? (how many replicate?) Maybe it is because I didn't understand, but how can particles bigger than 1 mm ( figure 1) be internalized by cells smaller than 200 um (figure 2)? There are several problems with regards the representation of the statistics in most of the figures. Example figure 2b) Or you say in the captions what you are comparing or you show in the graphic. I really do think the authors should be careful in reporting their findings.

Author Response

Pharmaceuticals-1951892: Reviewer responses round 2

REVIEWER 1:

Overall response: Thank you again for your constructive feedback and time. We really appreciate it.

Comment: Based on the revised version, I suggest the acceptance of this paper. At the same time, I still recommend some very related papers should be cited, including https://www.sciencedirect.com/science/article/abs/pii/S0927775722008172

Response: Thank you for your advice. We have cited the paper at the end of the introduction, please see line 278. We have also added a couple more citations to the discussion and introduction.

Reviewer 2 Report

Based on the revised version, I suggest the acceptance of this paper. At the same time, I still recommend some very related papers should be cited, including https://www.sciencedirect.com/science/article/abs/pii/S0927775722008172; 

Author Response

Pharmaceuticals-1951892: Reviewer responses round 2

REVIEWER 1:

Overall response: Thank you again for your constructive feedback and time. We really appreciate it.

Comment: Based on the revised version, I suggest the acceptance of this paper. At the same time, I still recommend some very related papers should be cited, including https://www.sciencedirect.com/science/article/abs/pii/S0927775722008172

Response: Thank you for your advice. We have cited the paper at the end of the introduction, please see line 278. We have also added a couple more citations to the discussion and introduction.

REVIEWER 2:

Overall response: Thank you again for your constructive feedback and time. We hope you find the following improvements satisfactory. 

Comment: My only concern is regarding the captions of the figures. They should have more description or even the figures that they describe E.g. Figure 1. Brightfield images of swollen microparticles at 4x, the bar represent 1 mm. a) alginate b) alginate coated with chitosan c-d) SEM image of dried microparticles. e) Zeta potential of the BV and MPs. Where is the zeta image? Description for the zeta N=? (how many replicate?) E.g. Table 1. Microparticle properties. What is discribed in the table? Zeta, hydrodynamic size (by DLS?) N=? (how many replicate?)

Response: Thank you for your advice. We have included further detail into all figure captions, under Table 1, and to the methods section. The average zeta potential (from 3 replicates each with 10 runs) is presented in Table 1 and we removed the caption for the zeta image. Also, the methods for the zeta potential measurements are under section 4.9, line 1021.

Comment: Maybe it is because I didn't understand, but how can particles bigger than 1 mm ( figure 1) be internalized by cells smaller than 200 um (figure 2)?

Response: The spherical hydrogel elutes the baculovirus over time so it does not enter any of the cells. After the baculovirus is eluted, it binds to cells and enters them via endocytosis (as described by previous studies). Once inside the cell the VEGFA gene is expressed from the baculovirus.

Comment: There are several problems with regards the representation of the statistics in most of the figures. Example figure 2b) Or you say in the captions what you are comparing or you show in the graphic. I really do think the authors should be careful in reporting their findings.

Response: Thank you for your comment. We have revised all figure captions to show that all statistical analysis compares the treatments to the mock, unless otherwise stated. The mean and standard deviation is presented in each graph and this information is included in the methods section under “statistical analysis”.